# Coenzyme Q10 and Immune Function: An Overview

**DOI:** 10.3390/antiox10050759

**Published:** 2021-05-11

**Authors:** David Mantle, Robert A. Heaton, Iain P. Hargreaves

**Affiliations:** 1Pharma Nord (UK) Ltd., Newcastle NE61 2DB, UK; dmantle@pharmanord.com; 2School of Pharmacy, Liverpool John Moores University, Liverpool L3 3AF, UK; R.Heaton@2013.ljmu.ac.uk

**Keywords:** coenzyme Q10, mitochondria, lysosome, inflammation, reactive oxygen species, oxidative stress

## Abstract

Coenzyme Q10 (CoQ10) has a number of important roles in the cell that are required for optimal functioning of the immune system. These include its essential role as an electron carrier in the mitochondrial respiratory chain, enabling the process of oxidative phosphorylation to occur with the concomitant production of ATP, together with its role as a potential lipid-soluble antioxidant, protecting the cell against free radical-induced oxidation. Furthermore, CoQ10 has also been reported to have an anti-inflammatory role via its ability to repress inflammatory gene expression. Recently, CoQ10 has also been reported to play an important function within the lysosome, an organelle central to the immune response. In view of the differing roles CoQ10 plays in the immune system, together with the reported ability of CoQ10 supplementation to improve the functioning of this system, the aim of this article is to review the current literature available on both the role of CoQ10 in human immune function and the effect of CoQ10 supplementation on this system.

## 1. Introduction

When the benefits of coenzyme Q10 (CoQ10) supplementation are being promoted, activation of the immune system is invariably quoted; however, such statements are rarely supported by documentary evidence referenced from the peer-reviewed medical literature. The objective of this article is therefore to review the current evidence in the literature for the role of CoQ10 in human immune function, including the benefits of CoQ10 supplementation in this new area of therapeutic utilization for CoQ10.

In order to determine the potential role of CoQ10 in immune function in a rational manner, one must first consider how the human immune system operates. In simplistic terms, the immune system is comprised of two parts, the innate immune system, and the adaptive immune system. As the name implies, the innate immune system is present and operational from birth, providing a rapid and non-specific first-line response to invading microorganisms. The innate immune system comprises a number of components, the most prominent being phagocytic cells (macrophages, neutrophils, natural killer cells) which destroy engulfed invading microorganisms via the generation of free radical species (reactive oxygen species (ROS) and reactive nitrogen species (RNS)). Subsequent to the innate response, the adaptive immune system may be activated, depending on the severity of infection. Adaptive immunity is a slower responding but more specific form of immune defence, involving B and T lymphocytes. B lymphocytes produce antibodies to neutralise specific antigens, whereas T lymphocytes have a role in destroying infected host cells. Communication between the various cell types of the two branches of the immune system is facilitated via cytokine protein chemical messenger molecules.

There is a common misconception that inflammation, which involves the release of pro-inflammatory cytokines, is a wholly negative process within the body. However, inflammation is the body’s normal response to infection or injury, and is essential for tissue healing, although this process should resolve following the initial immune response [1]. This resolution occurs via negative feedback mechanisms involving the generation of specific lipid molecules such as resolvins, protectins, and maresins which are synthesized from essential omega-3 and omega-6, fatty acids, respectively [2]. A balance must therefore be achieved in immune defence by neutralising infectious organisms, without precipitating their so-called cytokine storm, the uncontrolled release and excessive release by the innate immune system of pro-inflammatory cytokines resulting in tissue injury.

CoQ10 performs a number of cellular functions of potential relevance to the immune system. Firstly, CoQ10 has a key role in cellular energy supply, via its essential role in oxidative phosphorylation within mitochondria. The immune response has intensive energy requirements, and an adequate supply of CoQ10 is therefore required to enable the various cell types of the immune system to function optimally. CoQ10 serves as an electron carrier within the mitochondrial respiratory chain (MRC), where it transports electrons derived from complex I and II to complex III, enabling a continuous supply of electrons that are required for the process of oxidative phosphorylation with the concomitant product of ATP [3] (Figure 1). Secondly, CoQ10 also functions as an important lipid-soluble antioxidant, protecting cellular membranes and circulatory lipoproteins from free radical-induced oxidative damage [4]. The antioxidant function of CoQ10 is attributed to its fully reduced ubiquinol form [3]. Since phagocytic cells destroy invading pathogens via the production of free radicals [5], the antioxidant action of CoQ10 may protect phagocytic cells from self-destruction caused by their generation of free radicals. Finally, CoQ10 is able to directly modulate the action of genes involved in inflammation and may have a role in controlling the release of pro-inflammatory cytokines in disorders where this may be required [6]. The objective of the present article is therefore to review evidence from the Medline literature database for the role of CoQ10 in any of the above functions relevant to the human immune response.

## 2. CoQ10 and Animal Models of Immune Function

Immunosenescence is defined as the gradual deterioration of immune function associated with the aging process and is one of the major factors contributing to morbidity and mortality in the elderly [7]. In general terms, immunosenescence is characterised in a variety of species including the loss of function of most immune system cell types, including B-cells, T-cells, and NK (natural killer)-cells, with a corresponding increase in susceptibility to infection and cancer. The phenomenon of immunosenescence has been studied extensively in animal models, particularly the mouse [8,9]. As noted above, ageing in mice is associated with a loss of function of B-cell and T-cell function [10,11]. In aged mice (22 months) compared to young mice (10 weeks), Bliznakov [12] demonstrated a suppression of immune function associated with a deficiency of circulatory CoQ10 status; immunological function was in part restored following administration of CoQ10 [12,13]. In this study, CF1 mice were injected with sheep red blood cells as antigen; the humoral haemolytic primary response of the aged mice was less than 50% of that of the younger mice. This depressed immune response in older mice was restored to approximately 80% of that seen in younger mice following intravenous administration of 125 mcg CoQ10 per mouse, via improved performance efficiency of B-cells and T-cells. In senescence-accelerated mice, dietary supplementation with CoQ10 in the form of ubiquinol increased levels of the regulatory proteins, sirtuins as well as the expression of PGC-1alpha (peroxisome proliferator-activated receptor coactivator 1 alpha) [14], which in turn mediate immune function [15].

In a mouse model of rheumatoid arthritis, administration of CoQ10 decreased serum immunoglobulin levels, reduced markers of oxidative stress and inflammation, and mitigated arthritis severity [16]. Several mouse models have been developed with deficiencies of five of the ten genes (designated *COQ1* to *COQ10*) involved in CoQ10 biosynthesis [17], although the link between individual gene deficiencies and immune function has, in general, yet to be established. However, a mouse model expressing a variant form of the monooxygenase enzyme, CoQ6 generated via the CRISPR-Cas9 gene-editing system showed increased susceptibility to infection and increased mortality following exposure to *S. pneumoniae*; this was in turn rationalised in terms of impaired macrophage function, with reduced mitochondrial activity and an intrinsic inability to destroy internalised bacteria [18]. However, in this study, no assessment was made of endogenous CoQ10 status or mitochondrial function and therefore, further studies are required to confirm or refute this supposition. As alveolar macrophages are the first responders in the lung to bacterial challenge, the inability of these macrophages to mount a sufficient immune response can explain the observed increase in mortality following bacterial pneumonia. Additionally of relevance in this regard is a study in the fruit fly (*Drosophila melanogaster*), in which mutants with deficient levels of the gene, *COQ2* showed increased susceptibility to infection by bacteria and fungi, which was in part reversed following supplementation with CoQ10 [19].

Using SPF (specific pathogen-free) mice, administration of CoQ10 (0.5 g/Kg) resulted in increased production of T-cells and increased macrophage phagocytic capacity [20]. In mice with virally induced myocarditis, administration of CoQ10 resulted in reduced tissue inflammation and improved survival rate to infection [21].

The major form (approximately 95%) of coenzyme Q in humans is CoQ10, with less than 5% of the total coenzyme Q present as coenzyme Q9 (CoQ9; [22]). However, in rodents, the major form of coenzyme Q in tissues is present as CoQ9, with CoQ10 present in lesser amounts. The question, therefore, arises as to whether supplementation with CoQ9 can mediate immune function in mice or rats. However, there is little data in the literature to answer this question. Novoselova et al. [23] reported that suppression of B-cell and T-cell immune response in mice following irradiation could be partially restored following dietary supplementation with CoQ9. It is of note that dietary supplementation with CoQ10 is able to increase both CoQ9 and CoQ10 levels in mice indicating the ability of these animals to demethylate the isoprenoid side chain of CoQ10 [24].

## 3. CoQ10 and Susceptibility to Infection

Several clinical studies have linked depleted CoQ10 levels to an increased susceptibility to infection. Thus, Chase et al. [25] reported significantly reduced serum CoQ10 levels in patients with influenza compared to healthy control subjects. In children hospitalised with pandemic influenza (H1N1), Kelekçi et al. [26] reported a significant correlation between depletion of serum CoQ10 levels and chest radiographic findings. In a randomised placebo-controlled clinical trial, elderly patients with pneumonia showed significantly improved recovery following administration of CoQ10 (200 mg/day for 14 days) compared to the placebo group with a shortening of the symptomatic period and duration of antibiotic treatment being reported [27]. Unfortunately, no assessment of circulatory CoQ10 status was undertaken in this study and therefore the therapeutic plasma/serum level of this quinone that was eliciting a beneficial effect to patients could not be determined. 

Specifically with regard to infection with SARS-CoV-2 virus, in a clinical study by Israel et al. [28], intake of CoQ10 was associated with a significantly reduced risk of hospitalisation from SARS-CoV-2. In this large population study, patients hospitalised following SARS-CoV-2 infection were assigned to two case-control cohorts, which differed in the manner in which control subjects were selected—either from the general population or from patients infected with SARS-CoV-2 but not requiring hospitalisation. From a range of substances investigated, three were identified which significantly reduced the risk of hospitalisation following SARS-CoV-2 infection, most notably the ubiquinone form of CoQ10 (odds ratio 0.185, 95% confidence interval, *p* < 0.001), together with ezetimibe (inhibits the intestinal absorption of cholesterol) and the statin, rosuvastatin, a competitive inhibitor of the enzyme, HMG-CoA reductase—all substances linked to the cholesterol synthesis pathway. Since RNA viruses such as SARS-CoV-2 are known to require cholesterol both to enter cells and for viral replication, the authors of this study considered the possibility that supplemental CoQ10 prevents the virus from hijacking the mevalonate pathway to produce cholesterol. Ayala et al. [29] reviewed evidence for mitochondrial dysfunction as a key factor determining the severity of SARS-CoV-2 infection; in particular, the authors noted the increased susceptibility to SARS-CoV-2 infection in individuals over 65 years of age, the same age by which levels of endogenous CoQ10 has become substantially depleted. Similarly, Gvozdjakova et al. [30] considered one of the main consequences of SARS-CoV-2 infection to be virus-induced oxidative stress (an imbalance between free radical generation and antioxidant defences) causing mutations in one or more of the genes responsible for CoQ10 biosynthesis, in turn resulting in mitochondrial dysfunction. A number of factors may contribute to RNA virus-induced oxidative stress including inflammation and virus-induced mitochondrial dysfunction [30]. Additionally of note is the computational study by Caruso et al. [31], in which the authors identified CoQ10 as a compound capable of inhibiting the SARS-CoV-2 virus, via binding to the active site of the main viral protease (SARS-CoV-2 Mpro protease) which is required for viral replication. In SARS-CoV-2 infections, a balance must be achieved in immune defence against the virus, without precipitating the so-called cytokine storm, the uncontrolled release of pro-inflammatory cytokines responsible for lung injury and respiratory distress in severely affected patients [32]. Folkers and colleagues have reported the ability of CoQ10 monotherapy as well as CoQ10 taken together with vitamin B_6_ (pyridoxine) to significantly increase the levels of T4-lymphocytes together with the immunoglobulin, IgG in human subjects providing further support for the use of this isoprenoid in the treatment of infectious diseases [33].

## 4. CoQ10 and Immune Function in Athletes

Whilst regular sessions of relatively short-lasting moderate-intensity exercise are beneficial to immune function, individuals undergoing intensive or prolonged exercise are subject to depression of immune function, and are more susceptible to infections, particularly infection of the upper respiratory tract. A number of immune parameters may be affected, including levels of immunoglobulins, cytokines, NK cell activity, and macrophage phagocytic activity, as well as inflammation [34]. A number of clinical studies have reported supplementation with CoQ10 can improve immune function in such individuals. Thus, Emami [35] found supplementation with CoQ10 for 14 days prevented adverse changes in the levels of pro-inflammatory cytokines in elite swimmers. In kendo athletes, supplementation with CoQ10 (300 mg/day for 20 days) modified sub-populations of monocytes associated with inflammation [36]. In junior athletes, supplementation with CoQ10 (60 mg/day for 28 days) resulted in reduced levels of pro-inflammatory cytokines [37]. However, in the study by Trushina et al. (2019) CoQ10 was administered together with carnitine and therefore the anti-inflammatory effect observed cannot be solely attributed to CoQ10.

## 5. CoQ10 and Immune Cell Activation

There is relatively little data available from clinical studies demonstrating direct activation of components of the innate or adaptive immune systems by CoQ10. In patients with type I diabetes, supplementation with CoQ10 (100 mg twice daily for 3 months) was found to improve NK activity compared to placebo; specifically, the activating receptor NKG2D on NK cells from patients was up regulated, and the proportion of CD56bright NK cells increased [38]

In healthy volunteers, Folkers et al. [33] reported supplementation with CoQ10 (200 mg/day for 2 months) resulted in a significant increase in the blood levels of a T-lymphocyte subtype (T4, responsible for immune response regulation) and IgG (the most common type of antibody produced by B-lymphocytes). However, since vitamin B6 (pyridoxine) was also included in the treatment protocol of this study, the beneficial effects of pyridoxine on immune function should also be taken into account [33]. 

In a randomised controlled study by Barbieri et al. [39], individuals undergoing vaccination for hepatitis B were supplemented with CoQ10 (180 mg/day for 90 days) and were found to have a significant increase (by 57%) in antibody response to hepatitis B surface antigen, compared to the placebo.

In a case report by Farough et al. [40], a 4-year-old child with immune dysfunction (manifested as abnormal T-cell function and frequent recurrent infections) was found to be CoQ10 deficient (via muscle biopsy analysis). Supplementation with CoQ10 (150 mg/day for 12 months) resulted in a significant improvement in T-cell function as measured by the proliferative response with interleukins and reduced incidence of infections. The improvement in immune function was accompanied by a plasma CoQ10 level above the reference range at 3 months and within the reference range at 12 months following supplementation. The authors suggested that patients with recurrent infections and functional T-cell deficiency should be investigated for CoQ10 deficiency and subsequent supplementation. The impaired immune function of the patient was attributed to a diminution in leukocyte cellular energy generation as the result of an impairment in CoQ10 biosynthesis.

## 6. CoQ10 and Inflammation

As noted above, whilst inflammation occurs as part of the normal immune response against infection or injury, there are circumstances where the inflammatory response does not resolve appropriately; thus, chronic inflammation has been implicated in a number of autoimmune/degenerative disorders. There is considerable evidence from randomised controlled clinical studies that CoQ10 can mediate such inflammation, via effects on circulatory pro-inflammatory markers such as C-reactive protein (CRP), interleukins 1 and 8 (IL-1, IL-8), and tumour necrosis factor-alpha (TNF) [41]. Examples of the diverse disorders in which the levels of these markers were significantly reduced following supplementation with CoQ10 (typically 100–300 mg/day for 2–3 months) include chronic kidney disease [42], non-alcoholic fatty liver disease [43], cardiovascular disease [44], polycystic ovary syndrome [45], as well as ageing [46]. In a meta-analysis of 17 randomised controlled trials of disorders characterised by chronic inflammation, Fan et al. [47] reported significant lowering effects of CoQ10 on circulatory levels of the inflammatory markers, CRP, IL-6, and TNF-alpha. CoQ10 was suggested to elicit its anti-inflammatory effects by the reduction of NFkB (nuclear factor kappa B) gene expression [47]. Since NFkB can be activated by ROS, the ability of CoQ10 to neutralise these free radical species may make an important contribution to its ability to inhibit the activation of this transcription factor [47]. 

## 7. CoQ10, Mitochondria, and Immune Function

In addition to its role as an anti-inflammatory agent as outlined in the previous section, another major mechanism by which CoQ10 may influence immunity is via the role of mitochondria in the immune response. Mitochondria may mediate immune function at multiple levels. Firstly, activation of immune cells is an energy-intensive process dependent on mitochondrial energy supply. Secondly, in addition to its role in energy supply, the role of mitochondria in cell signalling pathways associated with the immune response has been increasingly recognised. Following infection, mitochondria are a source of so-called damage-associated molecular patterns (DAMPS), which are recognised by specific receptors (pattern recognition receptors) on the surface of macrophages, resulting in activation of the latter cells [48]. A second example of mitochondrial involvement in immune response cell signalling pathways involves the mitochondrial anti-virus signalling protein (MAVS protein), activation of which following an infection triggers cytokine release [49]. Thirdly, while free radical production during phagocytosis results primarily from the action of NAPDH oxidases (NOX1), free radicals such as ROS produced directly by mitochondria may also contribute to the destruction of invasive microorganisms [50]. Since CoQ10 is required for the normal functioning of mitochondria in the above processes, this provides another link between CoQ10 and immune function.

## 8. CoQ10, Lysosomes and Immune Function

Lysosomes have an important role to play in the immune system, where their hydrolytic enzyme activities are required for antigen processing and presentation [51], phagocytosis [52], and cytokine release [53]. In phagocytosis, lysosomes are critical for the maturation of phagosomes to phagolysosomes, which is required for cellular defence against pathogens; in macrophages, this process initiates a signalling cascade that connects the innate and adaptive immunity systems, enabling a continuous immune response [54]. The lysosome contains some seventy hydrolase enzymes requiring an acidic environment of between pH from 4.5 to 5.1 for optimum activity, and even a small increase in pH has been reported to inhibit the activity of these enzymes [55]. Thus, the pH of the lysosomal lumen is intrinsically linked to the function of this organelle.

The acidic environment of the lysosomal lumen is maintained predominantly by a V-ATPase, which uses the free energy liberated from the hydrolysis of ATP to pump protons from the pH neutral (7.2) cytosol across the lysosomal membrane and into the lumen of the organelle [49]. The majority of the ATP required for this process is provided by MRC oxidative phosphorylation, for which CoQ10 serves as an important electron carrier [3]. CoQ10 can also be found in the lysosomal membrane, where it functions as both an electron and proton carrier in the tentative respiratory chain of this organelle (LRC), which also functions to maintain the acidic environment of the lumen [56]. A recent study by Heaton et al. [57] reported an increase in lysosomal pH from 5.1 to 6.2 following a pharmacologically induced CoQ10 deficiency in neuronal cells, confirming the important role of this isoprenoid in maintaining the acidity of the lysosome. Interestingly, lysosomal storage disorders, which are inherited genetic disorders that impair the function of this organelle, have been associated with aberrations in the function of the immune system, ranging from immunosuppression in disorders such as Gaucher’s disease to an enhanced or autoimmune response in the disease, juvenile neuronal ceroid lipofuscinosis [53].

During the current pandemic, a study by Ghosh et al. [58] has emerged that reports the ability of SARS-CoV-2 to utilise a lysosomal dependent exocytosis pathway for their release into the extracellular environment from the host cell, resulting in viral spreading. However, as a consequence of the viral exploitation of this process, lysosomes become de-acidified with the consequent impairment of hydrolase enzyme activity and disrupt antigen presentation pathways. The authors speculate that targeting regulators of lysosomal trafficking, biogenesis, or reversing the de-acidification of the lysosome may enhance the immune response against the virus and may mitigate coronavirus infections [58]. In view of the essential role CoQ10 plays in maintaining lysosomal acidification both as an electron carrier and/or proton carrier in the MRC and LRC respectively, CoQ10 supplementation may be an appropriate therapeutic strategy to consider in pre and post COVID-19 treatment.

## 9. CoQ10, Peroxisomes and Immune Function

In addition to their role in oxidative catabolism, the role of peroxisomes in the immune response has also been recognised [59]. Peroxisomes initiate signalling of the presence of an intracellular virus, thereby leading to the first rapid response to an infection via interferon production [60]. Free radical species generated within peroxisomes such as superoxide may also contribute to the destruction of invasive microorganisms during phagocytosis. As a component of peroxisomal membranes, CoQ10 may help to maintain organelle integrity by protecting the membranes against free radical-induced oxidative damage [61]. Indeed, peroxisomes are thought to be involved in the biosynthesis of CoQ10 and have been reported to contain two of the enzymes involved in the synthesis of this isoprenoid [62].

## 10. CoQ10 and Immune Function in Cancer

Evidence of decreased plasma concentrations of CoQ10 has been reported in patients with cancer (breast cancer, myeloma, lymphoma, and lung cancer) [63,64]. Furthermore, Jolliet et al. [65] reported decreased levels of plasma CoQ10 in both patients with breast cancer, and also in patients with non-malignant breast disease. These results indicated that the decreased CoQ10 levels may also be responsible for benign mammary cell growth. The study also found a statistically significant relationship between the plasma CoQ10 level and breast cancer prognosis. However, a subsequent study in prostate cancer found no association between circulatory CoQ10 status and the risk of developing the disease, although an optimal CoQ10 plasma level of >9.3 nmol/L was suggested as a requirement to reduce cancer risk [66]. Interestingly, a study in patients with common variable immunodeficiency, a disease in which is frequently associated with cancer found evidence of a significant decrease in plasma ubiquinol levels compared to age-matched controls, although there was only a slight loss of plasma total antioxidant capacity. The loss of CoQ10 status in these patients may contribute to the high level of inflammatory mediators reported in this disease [67]. The decreased circulatory CoQ10 levels reported in cancer patients may be due to increased consumption of this molecule by ROS or as the result of increased metabolic demand by the tumour cells [68]. In view of the strong association between inflammation and cancer, the deficit in CoQ10 status reported in cancer patients may have important implications for tumorigenesis [69]. The overexpression of the inflammatory cytokines, IL-6 and TNF have been reported in the tumour microenvironment, where they promote all the hallmarks of cancer, including cell proliferation, angiogenesis, invasiveness, and metastasis [69]. Interestingly, since CoQ10 has been reported to decrease the circulatory level of these cytokines [47], the deficit in CoQ10 status reported in cancer patients may contribute to the high levels of IL-6 and TNF detected in this disease [69,70]. Indeed, a preliminary study in six cancer patients reported the ability of CoQ10 therapy to reduce the circulatory levels of TNF, although no details were provided as to whether this was associated with any regression of the tumour [71]. Although the use of CoQ10 in the treatment of cancer is still in its infancy, other studies have reported evidence of remission following supplementation [72,73]. It is believed that the anti-inflammatory actions of CoQ10 in cancer may be mediated by the inhibition of the activation of transcription factors NFkB (nuclear factor kappa B)/AP-1(activator protein-1), which as well as being involved in multiple aspects of innate and adaptive immune function also appears to have an important role in carcinogenesis. Furthermore, CoQ10 may also induce the expression of the transcription factors Nrf2 (nuclear factor erythroid 2-related factor 2), which binds to the antioxidant response element that activates a battery of genes resulting in the expression of antioxidant and detoxifications proteins, in turn resulting in a suppression of the inflammatory response [74]. A study by Vetvicka et al. [75] indicated the synergistic effect of combined CoQ10 and beta-glucan therapy in the treatment of cancer. Beta-glucan is a glucose polymer with widely described immunomodulatory properties, which in the case of cancer involves activation of immune cells (either directly or via cytokine induction) that are able to directly eradicate the tumour [75]. However, co-treatment of murine breast and lung models with CoQ10 and beta-glucan was found to induce a more pronounced loss of breast tumour volume or number of metastases respectively than either treatment in isolation, although the actual mechanism responsible for therapeutic synergy was not investigated [76].

CoQ10 has been reported to offer some protection against the cardiotoxic side-effects of the anthracycline antibiotic drug adriamycin, which is commonly used as a chemotherapy agent [76,77]. Although the antioxidant capacity of CoQ10 appears to be integral to its ability to protect the heart from adriamycin toxicity [78], its anti-inflammatory function has also been reported to play a central role in attenuating radiotherapy-associated cellular damage. In a study by Yakin et al. [79], it was found that radioiodine treatment caused pronounced inflammatory response in the lacrimal glands (exocrine glands located above the eyeball); this was indicated by oxidative damage to the tissue, decreased total antioxidant status, and increased levels of the pro-inflammatory cytokines TNF and IL-6. CoQ10 treatment increased the total antioxidant status as well as decreasing the levels of TNF and IL-6. Interestingly, the level of total oxidative stress detected in the lacrimal glands following radioiodine treatment was not lowered by CoQ10 supplementation. A subsequent study by Mohamed and Said [80] investigated the potential of CoQ10 treatment to protect against radiotherapy-induced enteropathy in rats. Following exposure to ionising radiation, evidence of cell membrane damage and lipid peroxidation, together with decreased levels of the cellular antioxidants reduced glutathione GSH and the antioxidant enzyme catalase, were detected in the same intestine. In addition, the radiation increased mucosal neutrophil infiltration and up-regulated the levels of the pro-inflammatory cytokines IL-6 and cyclooxygenase 2 (COX-2). CoQ10 treatment was found to decrease the production of IL-6 and COX-2 in the small intestine, as well as increase both the level of GSH and the activity of catalase, in addition to decreasing the level of lipid peroxidation as indicated by malondialdehyde [80]. Interestingly, CoQ10 treatment was found to decrease the expression of intestinal NfkB, which was thought to be directly responsible for the decrease in the levels of IL-6 and COX-2. Although not investigated, the ability of CoQ10 to induce Nrf2 may explain its effect on the antioxidant status of the intestinal tissue.

Cancer cells have been known for a long time to be able to tame or dampen down the immune system preventing an immune response from being directed at the cancer. However, a CoQ10 derivative known as 4-aetylantroquinonol derived from the mycelium of Antrodia cinnamonea has the potential to improve the anti-tumour immune response, by increasing the antigen-presenting ability of dendritic cells and their ability to secrete immune-related cytokines decreasing the secretion of immune escape related cytokines, IL-6 and IL-10 [81].

## 11. Coenzyme Q10 Monitoring and Dosage

In general, clinical monitoring of CoQ10 status is based on plasma determinations, with an established reference interval ranging from 0.5 to 1.7 μM [82]. The actual reference range can vary between centres, possibly attributable to the different methods used to determine CoQ10 levels. Generally, HPLC linked to either a UV or electrochemical detector is the method used to determine CoQ10 status, although mass spectrometry is becoming more readily utilised for this determination [82]. Since the level of plasma CoQ10 is influenced by both diet and circulatory lipoprotein status, it may be an inappropriate surrogate to assess endogenous CoQ10 status, blood mononuclear cells or urine epithelial cells may be more appropriate for this determination [83]. However, the “gold standard” for the assessment of tissue CoQ10 status is through skeletal muscle, although this is a relatively invasive surrogate. Currently, there is no consensus on the dosage of CoQ10 or the plasma level required that may prove efficacious in optimising the immune function or mediating chronic inflammation. Coenzyme Q10 supplementation is safe and well-tolerated, exhibiting an excellent safety profile, with doses as high as 2400 mg/d being used in the treatment of Parkinson’s disease [84]. Typically, doses in the range of 5 to 50 mg/kg/d have been administered to patients with documented low levels of tissue CoQ10 or mitochondrial disease [83], although there is limited data on the therapeutic doses required to modulate the immune system. Doses ranging from 10–30 mg/kg/day have shown some efficacy in rat studies, although the doses used in humans have varied considerably, with few taking into account human body weight [79,80]. Furthermore, no therapeutic target plasma levels of CoQ10 have so far been determined for optimal immunomodulatory activity. In Parkinson disease, a plasma CoQ10 level of 4.6 μmol/L was reported to be the most effective in slowing functional decline in patients, although this therapeutic effect was thought to be related more to the antioxidant and mitochondrial functioning properties of CoQ10, rather than to any effect on the immune system [84].

## 12. Safety and Bioavailability of CoQ10

CoQ10 is generally well tolerated, with no serious adverse effects reported in long-term use. Very rarely, individuals may experience mild gastrointestinal disturbance. There are no known toxic effects, and CoQ10 cannot be overdosed [85]. The safety of CoQ10 has been confirmed in more than 200 randomised controlled trials, on a wide range of disorders. A recent clinical study by López-Lluch et al. [86] demonstrated the variation of quality and bioavailability of CoQ10 supplements, highlighting the need to utilise a product manufactured to pharmaceutical standards. The bioavailability of seven different supplement formulations containing 100 mg of CoQ10 was evaluated in 14 young healthy individuals. Bioavailability was measured as the area under the curve (AUC) of plasma CoQ10 levels over 48 h after the ingestion of a single dose. Measurements were repeated in the same group of 14 volunteers in a double-blind crossover design with a minimum of a four-week washout between intakes. The bioavailability of the formulations showed large differences that were statistically significant. The best absorbable formulation was a soft-gel capsule containing ubiquinone (Bio-Quinone Q10). The matrix used to dissolve CoQ10 and the proportion and addition of preservatives such as vitamin C affected the bioavailability of CoQ10. In particular, the importance of CoQ10 crystal dispersion in the initial formulation is emphasised, the absence of which reduces the bioavailability by 75% of what it would be if included.

The efficiency of the small intestine to absorb CoQ10 appears to decrease as the dosage of the isoprenoid increases, with a suggested block in absorption above 2400 mg [82]. Rather than one large dose of CoQ10, split doses have been recommended and dietary fat, together with grapefruit juice, have been reported to improve the absorption of CoQ10 [87,88]. In contrast, ingestion of high-dose vitamin E together with CoQ10 may impede the absorption of the isoprenoid resulting in lower circulatory levels of CoQ10, possibly as a result of competition during the small intestinal absorption [89].

## 13. Discussion

CoQ10 is required for the optimal function of the immune system, as well as mediating inflammatory response in disease. The functions of CoQ10 supplementation in the treatment of immunopathy and in the immune function of other diseases are outlined in Figure 2. Indeed, the prospect of immune dysfunction has been associated with human CoQ10 deficiency [40]. In view of the ability of CoQ10 supplementation to enhance the activity of the immune cells, especially the B and T lymphocytes [14], as well as ameliorate the inflammatory response [47], it may be appropriate for use in a number of diseases of the immune system. One such disease is the neurodegenerative disorder multiple sclerosis (MS), an immune-mediated disease of the central nervous system. A study by Sanoobar et al. [90] reported the ability of CoQ10 supplementation to reduce the circulatory levels of the inflammatory markers (TNF, IL-6, and metallopeptidase 9; MMP-9) in MS patients. However, no assessment was undertaken on the cerebral inflammatory response following CoQ10 supplementation. At present, it is uncertain whether CoQ10 is able to cross the blood-brain barrier (BBB) of humans, and a study by Wainwright et al. [91] has added further support to this supposition. However, in this study, it was suggested that uptake of exogenous CoQ10 into the brain may be enhanced by the administration of LDLR inhibitors, or by interventions to stimulate the luminal activity of the Scavenger Receptor, SR-B1 transporters. Furthermore, the use of analogues of CoQ10 such as idebenone or MitoQ, which have the ability to cross the BBB, may also be judicious in the treatment of neuro-immune dysfunction, although these synthetic quinones may not have the same beneficial effect on the immune system as CoQ10 [90]. The CoQ10 status of individuals may also be an important determining factor in susceptibility to COVID-19 [28] in view of the suggested ability of CoQ10 to impair virus replication [30], ameliorate oxidative stress [3], modulate the inflammatory response [47], and optimize lysosomal function [57]. Although, the ability of CoQ10 to enhance the mitochondrial function of leukocytes may also be an important factor to consider [40]. 

Supplementation with CoQ10 can be a cost-effective treatment for a variety of disorders [3]. For example, 150 capsules of Bio-Quinone 100 mg CoQ10, a food supplement manufactured to pharmaceutical standards, currently retails at £76 in the UK, equivalent to a cost of 50 pence per capsule; with a recommended dosage typically in the range of 100–200 mg per day, this equates to a treatment cost of 50 pence to £1 per day. It should be noted that the majority of CoQ10 products are sold as food supplements, which in contrast to licensed pharmaceutical products, can vary considerably in both quality and price; only one CoQ10 product (Myoquinon, the licensed version of Bio-Quinone) currently has a marketing authorisation within the European Commission for the adjunctive treatment of heart failure.

Overall, further studies are required to elucidate the optimal doses and plasma levels of CoQ10 required to optimize immune function and mediate inflammation, so that appropriate treatment protocols can be established.

## 14. Conclusions

Administration of supplemental CoQ10 has been shown (on the basis of randomized controlled clinical trials) to benefit a number of disorders, especially the prevention [92] or treatment [93] of cardiovascular disease, but also diabetes [94], non-alcoholic fatty liver disease [95], and chronic kidney disease [96], as well as some neurological disorders [97]. In particular, the use of supplemental CoQ10 for the treatment of heart failure has become well-established based on such evidence. However, when activation of the immune system is quoted as a benefit of CoQ10 supplementation, such statements are rarely supported by documentary evidence referenced from the peer-reviewed medical literature, thus necessitating a review of the type given in the present article. Relatively little data are available from clinical studies demonstrating direct activation of components of the innate or adaptive immune systems by CoQ10, and further studies in this area would be of value. However, CoQ10 has a number of characteristics of relevance to immune function; whilst the anti-inflammatory role of CoQ10 has become increasingly recognized, the role of CoQ10 in lysosomal and peroxisomal function during the immune response represents areas that would benefit from further research.

## Figures and Tables

**Figure 1 antioxidants-10-00759-f001:**
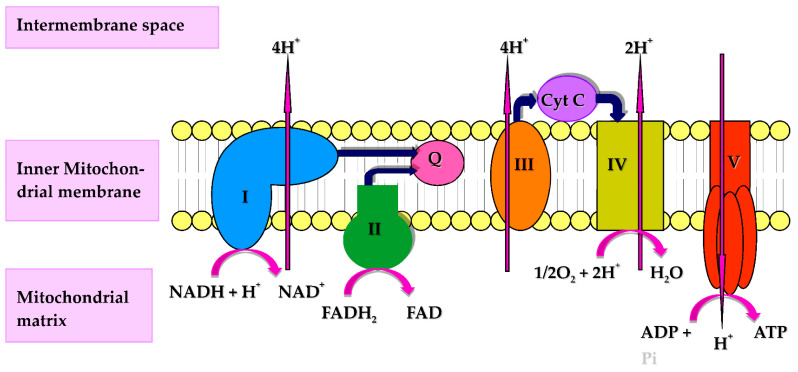
Diagram of the mitochondrial respiratory chain (MRC) and complex V illustrating proton (H^+^) movement during oxidative phosphorylation. Q: Coenzyme Q_10_; Cyt C: Cytochrome c.

**Figure 2 antioxidants-10-00759-f002:**
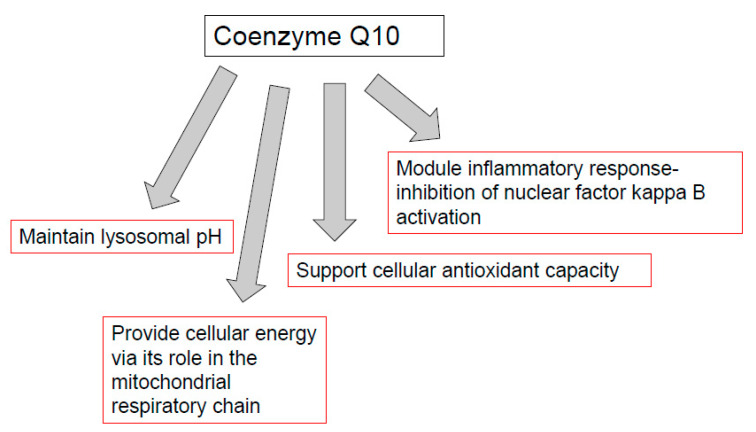
Functions of coenzyme Q10 supplementation in immunopathy, ageing, cancer, and immune function in athletes. TLR-4: toll-like receptor 4 (TLR-4). NFkB: nuclear factor kappa B; AP-1: Activator protein-1. PGC-1 alpha: peroxisome proliferator-activated receptor coactivator 1 alpha. Nuclear factor erythroid 2-related factor 2.

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
