# Peer review of "Coenzyme Q10 and Immune Function: An Overview"

_antioxidants, 2021, doi:10.3390/antiox10050759_

Round 1

Reviewer 1 Report

This review provides important information from history CoQ10 to the present. Minor corrections are required:

1./ References, page 3. Rouseau et al 1998 - is missing in references

2./ Reference No. 57 - is not complete.

3./ Authors reviewed the role and known functions of CoQ10 and its supplementation in human diseases, a relative new is part - its immune functions.

4./ Page 4: authors used....infection with COVID-19 virus (instead with SARS-CoV-2 virus)

5./ Page 10: Instead part 13. Discussion - will be better part: Conclusion and future directions.

Author Response

We thank the reviewer for taking the time to read our review and we have addressed the comments below:

1./ References, page 3. Rouseau et al 1998 - is missing in references-

We have included this reference in the amended review.  Details  are provided at  the end of reference list.

2./ Reference No. 57 - is not complete.

Missing part of reference  is now included in the reference list.

3./ Authors reviewed the role and known functions of CoQ10 and its supplementation in human diseases, a relative new is part - its immune functions.

Sorry, we are not sure about the meaning of this comment , but we have included the following sentence in the introduction- `including the benefits of CoQ10 supplementation in this new area of therapeutic utilization for CoQ10.`

4./ Page 4: authors used....infection with COVID-19 virus (instead with SARS-CoV-2 virus).

We thank the reviewer and have changed COVID-19 to SARS-CoV-2 in the text.

5./ Page 10: Instead part 13. Discussion - will be better part: Conclusion and future directions. A conclusion section has now been added to this review.

Reviewer 2 Report

I’ve read with attention the narrative review by Mantle et al. that is interesting, well-organized, overall well-written and update. The discussion should be improved and enriched including a mention on the proven positive effects of Q10 beyond the immune system and a mention of the cost of the treatment. Moreover, a table resuming the available clinical evidence could be useful for the reader.

Author Response

Referee-2

We thank the reviewer for their comments and have provided the following responses in yellow

I’ve read with attention the narrative review by Mantle et al. that is interesting, well-organized, overall well-written and update. The discussion should be improved and enriched including a mention on the proven positive effects of Q10 beyond the immune system and a mention of the cost of the treatment. Moreover, a table resuming the available clinical evidence could be useful for the reader.

We thank the reviewer and have included information of the benefits of supplementary CoQ10 in disorders other than immune function have been noted in the Conclusions section:

`Administration of supplemental CoQ10 has been shown (on the basis of randomized controlled clinical trials) to benefit a number of disorders, especially the prevention [92] or treatment [93] of cardiovascular disease, but also diabetes [94], non-alcoholic fatty liver disease [95] and chronic kidney disease [96], as well as some neurological disorders [97]. In particular the use of supplemental CoQ10 for the treatment of heart failure has become well-established based on such evidence. However, when activation of the immune system is quoted as a benefit of CoQ10 supplementation, such statements are rarely supported by documentary evidence referenced from the peer-reviewed medical literature, thus necessitating a review of the type given in the present article. Relatively little data is available from clinical studies demonstrating direct activation of components of the innate or adaptive immune systems by CoQ10, and further studies in this area would be of value. However, CoQ10 has a number of characteristics of relevance to immune function; whilst the anti-inflammatory role of CoQ10 has becoming increasingly recognised the role of CoQ10 in lysosomal and peroxisomal function during the immune response represent areas which would benefit from further research.`

 In addition we have included information on the cost of CoQ10 supplementation in the discussion:

`Supplementation with CoQ10 can be a cost-effective treatment for a variety of disorders [3]. For example, 150 capsules of Bio-Quinone 100mg CoQ10, a food supplement manufactured to pharmaceutical standards, currently retail at £76 in the UK, equivalent to a cost of 50 pence per capsule; with a recommended dosage typically in the range 100-200mg per day, this equates to a treatment cost of 50 pence to £1 per day. it should be noted that the majority of CoQ10 products are sold as food supplements, which in contrast to licensed pharmaceutical products, can vary considerably in both quality and price; only one CoQ10 product (Myoquinon, the licensed version of Bio-Quinone) currently has a marketing authorisation within the European Commission for the adjunctive treatment of heart failure.`

Reviewer 3 Report

The authors review the role of coenzyme Q10 in human health, focusing on the immune system function. The specific points include a description of mouse models, such as aging mice and models for autoimmune disease. Studies using humans involve elderly groups, athletes, and cancer patients, as well as susceptibility to infections.

One major point is the reference to unpublished data, end of page 4. Do the authors have the necessary permission to discuss these data? Does the journal allow including unpublished data in the review article?

The second point, Figure 2 can be updated using the text of the manuscript, including functions of CoQ10 in the aging population, in athletes, cancer patients, etc

Moreover, consider including the information on funding, conflict of interest, and acknowledgments

Author Response

Referee-3

The authors review the role of coenzyme Q10 in human health, focusing on the immune system function. The specific points include a description of mouse models, such as aging mice and models for autoimmune disease. Studies using humans involve elderly groups, athletes, and cancer patients, as well as susceptibility to infections.

One major point is the reference to unpublished data, end of page 4. Do the authors have the necessary permission to discuss these data? Does the journal allow including unpublished data in the review article?

We thank the reviewer for the comment. Permission to use this data was provided by Dr W. Judy in an e-mail to Dr Mantle dated 20th August 2020; a note to this effect has now  been included in the text.

The second point, Figure 2 can be updated using the text of the manuscript, including functions of CoQ10 in the aging population, in athletes, cancer patients, etc.

We thank the reviewer and have amended figure 2 accordingly to encompass the uses  of CoQ10 supplementation in immunopathy, cancer, athletes and aging.

Moreover, consider including the information on funding, conflict of interest, and acknowledgments.

We thank the reviewer and have provided this information in the amended paper.

No acknowledgement of funding or any other issues was required. With regards to possible conflict of interest, Dr Mantle is listed as medical adviser to Pharma Nord (UK) Ltd in the manuscript heading.

Round 2

Reviewer 3 Report

The authors revised the manuscript and provided answers to most of the questions.

Major point:

I am not satisfied with the description of unpublished data (my concern from revision 1). The authors refer to Dr. Judy's permission to publish these data, although it is not clear to me who is Dr. Judy, does she have any permission to share these data, and why she is not listed as a co-author if she contributed data used in the manuscript. I will let editors decide on this point, and I will no longer hold the publication of this review.

"Dr W. Judy, personal communication; permission to quote this data was granted by Dr Judy via e-mail to Dr Mantle on 20th August 2020)"

Minor points:

1. Figure 1. "Intermembrane" word is cut at the bottom, consider to fix it. 

"Mitochondri

al matrix". "Mitochondrial" could be in one line.

2. "This data" is not correct. Data is plural, thus it is "these data".

Author Response

Referee 3:

We thank the referee for taking the time to read through our review again and make suggestions. We have included the reviewer`s comments together with our answers below:

I am not satisfied with the description of unpublished data (my concern from revision 1). The authors refer to Dr. Judy's permission to publish these data, although it is not clear to me who is Dr. Judy, does she have any permission to share these data, and why she is not listed as a co-author if she contributed data used in the manuscript. I will let editors decide on this point, and I will no longer hold the publication of this review.

"Dr W. Judy, personal communication; permission to quote this data was granted by Dr Judy via e-mail to Dr Mantle on 20th August 2020)"

We agree with the reviewer and have removed this data from section 3 of the paper.

Minor points:

  1. Figure 1. "Intermembrane" word is cut at the bottom, consider to fix it. 

"Mitochondri

al matrix". "Mitochondrial" could be in one line.

We apologise and have now amended Figure 1 so `Mitochondrial` is all on the same line in the figure.

  1. "This data" is not correct. Data is plural, thus it is "these data".

We have now removed the data and this comment from the review.